



# Possible expression of the 4.2 kyr event in Madagascar and the south-east African monsoon

Nick Scroxton[1,2,3], Stephen J. Burns[1], David McGee[2], Laurie R. Godfrey[4], Lovasoa Ranivoharimanana[5], Peterson Faina[5]

[1]School of Earth Sciences, University College Dublin, Belfield, Dublin 4, Ireland
[2]Department of Geosciences, 611 North Pleasant Street, University of Massachusetts Amherst, MA 01030, USA
[3]Department of Earth, Atmospheric and Planetary Sciences, Massachusetts Institute of Technology, 77 Massachusetts Avenue, Cambridge, MA 02139, USA
[4]Department of Anthropology, 240 Hicks Way, University of Massachusetts, Amherst, MA 01003, USA
[5]Mention Bassins sédimentaires, Evolution, Conservation (BEC) – BP 906 – Faculté des Sciences, Université d'Antananarivo – 101 Antananarivo, Madagascar

*Correspondence to*: Nick Scroxton (nick.scroxton@ucd.ie)

**Abstract** The 4.2 kyr event is regarded as one of the largest and best documented abrupt climate disturbances of the Holocene. Drying across the Mediterranean and Middle East is well established and is linked to societal transitions in the Akkadian, Egyptian and Harappan civilizations. Yet the impacts of this regional drought are often extended to other regions and sometimes globally. In particular, the nature and spatial extent of the 4.2 kyr event in the tropics have not been established. Here, we present a new stalagmite stable isotope record from Anjohikely, northwest Madagascar. Growing between 5 and 2 kyr BP, stalagmite AK1 shows a hiatus between 4.32 and 3.83 kyr BP, replicating a hiatus in another stalagmite from nearby Anjohibe, and therefore indicating a significant drought around the time of the 4.2 kyr event. This result is the opposite to wet conditions at 8.2 kyr BP, suggesting fundamentally different forcing mechanisms. Elsewhere in the south-east African monsoon domain dry conditions are also recorded in sediment cores in Lake Malawi and Lake Masoko and the Taros Basin on Mauritius. However, at the peripheries of the monsoon domain, drying is not observed. At the northern (equatorial East Africa) and eastern (Rodrigues) peripheries, no notable event is record. At the southern periphery a wet event is recorded in stalagmites at Cold Air Cave and sediment cores at Lake Muzi and Mkhuze Delta. The spatial pattern is largely consistent with the modern rainfall anomaly pattern associated with weak Mozambique Channel Trough and a northerly austral summer Inter Tropical Convergence Zone position. Within age error, the observed peak climate anomalies are consistent with the 4.2kyr event. However, outside Madagascar, regional hydrological change is consistently earlier than a 4.26 kyr BP event onset. Gradual hydrological change frequently begins at 4.6 kyr BP, raising doubt as to whether any coherent regional hydrological change is merely coincident with the 4.2 kyr event rather than part of a global climatic anomaly.

## 1 Introduction

The recent formal subdivision of the Holocene (Walker et al., 2018; Walker et al., 2018) has proved controversial (Helama and Oinonen, 2019). In particular the middle to late Holocene (Northgrippian to Meghalayan) division defined at 4.20 kyr BP,



close to the onset of a significant Holocene climate anomaly occurred between 4.26 and 3.97 kyr BP, the so-called "4.2 kyr event". The 4.2kyr event is an abrupt climate anomaly between 4.26 and 3.97 kyr BP (Carolin et al., 2019), well documented

in the Mediterranean (Bini et al., 2019; Zanchetta et al., 2016) and Middle East (Kaniewski et al., 2018) as a widespread drought, contributing to societal change in the Akkadian civilization (Höflmayer, 2017; Weiss et al., 1993; Weiss, 1997; Höflmayer, 2017; Weiss et al., 1993; Weiss, 1997). However, both the spatial extent beyond the data-rich heartland of the northern hemisphere mid-latitudes, and the climate processes behind the 4.2 kyr event are uncertain. The 4.2 kyr event may be one of the smallest forced climate anomalies of the Holocene, perhaps through a freshwater input into the north Atlantic (Wang

et al., 2013; Wang et al., 2013), akin to a smaller version of the 8.2 kyr event. Alternatively, it may be one of the largest unforced (i.e. natural variability) climate anomalies of the Holocene (Yan and Liu, 2019), driven by changes in the North Atlantic Oscillation.

In particular, the impact of the 4.2 kyr event on the tropics and subtropics is unknown. The most often cited paper on the

subject is Marchant and Hoogiemstra (Marchant and Hooghiemstra, 2004), which provides a compilation of records from Africa and South America. This paper does not mention the 4.2 kyr event once, instead referring to a series of climatic changes around 4.0 kyr event, potentially associated with changing tropical sea-surface temperatures. This 4 kyr BP shift in tropical climate is now widely documented in the literature and likely related to changes in the mean state of ENSO (Denniston et al., 2013; Gagan et al., 2004; Giosan et al., 2018; Li et al., 2018; MacDonald, 2011; Toth and Aronson, 2019). No causal

relationship between the 4.0 kyr BP tropical climate shift and the 4.2kyr event has been established. Distinguishing between these two climatic events is crucial in understanding the spatial extent of the 4.2 kyr event.

An increasing number of tropical paleoclimate records now have the sampling resolution and dating precision to distinguish between the 4.2 kyr event and the 4.0kyr BP tropical climate shift. In this study we investigate the impacts of the 4.2 kyr event

on the south-west Indian Ocean monsoon domain. We present a new stalagmite $\delta^{18}O$ record of monsoon variability from north-west Madagascar, alongside other climate records from the region.

## 2 Climatology

The South-East African Monsoon (SEAfM), including the Malagasy Summer Monsoon (MSM), is driven by the annual southwards migration of the Inter-Tropical Convergence Zone during austral summer (Jury and Pathack, 1991; Jury et al.,

1995). The mountains of eastern Madagascar block the prevailing easterlies (Barimalala et al., 2018), allowing the cyclonic Mozambique Channel Trough (MCT) to form in the Mozambique Channel as the Mascarene High retreats to the south-west in the austral summer (Barimalala et al., 2020). As a result, between 30° and 50°E the summer rainfall band pushes down to 20°S, following the centre of convergence rather than peak regional sea-surface temperatures (SST) (Koseki and Bhatt, 2018),





but with local SST still playing a role in the mean state of the MCT (Barimalala et al., 2018). The rainfall band is almost

discontinuous from the rainfall band to the east and west beyond bounding meridional mountain ranges (Koseki and Bhatt, 2018). Winds originate from the Indian Winter Monsoon and Gulf of Oman moving south-west over the equatorial Indian Ocean before curving round to the south-east (i.e. northwesterlies) towards Madagascar. Moisture is likely derived from the equatorial west Indian Ocean and local sources (Scroxton et al., 2017).

Further south, moisture transport is driven by Tropical Temperate Troughs. Upper level mid-latitude baroclinic instabilities combined with low-latitude moist convection create a band of rainfall running north-west, south-east across southern Africa (Macron et al., 2014; Macron et al., 2014), with the Mature and Late Phases influencing rainfall on Madagascar (Macron et al., 2016). South of 25°S and the moisture blocking influence of the mountains of Madagascar, rainfall in southeast Africa is derived from the southeasterly trade winds, and while rainfall is still seasonal enough to considered monsoonal, there is no

seasonal wind reversal (Figure 1).

Interannual rainfall variability in northwest Madagascar is associated with changes in the strength of the MCT. Stronger cyclonic conditions lead to stronger westerlies in the Mozambique Channel towards Madagascar, greater onshore transport of moisture and increased rainfall. This pattern leads to a rainfall dipole between Madagascar and South Africa. A stronger MCT

is associated with a more southerly position of the Inter Tropical Convergence Zone (ITCZ) (Barimalala et al., 2020).

Unlike much of the circum-west Indian Ocean basin, tropical zonal atmospheric circulation variability such as the Indian Ocean Dipole plays a relatively weak role in MSM rainfall amount. The Indian Ocean Dipole is seasonally locked and, by definition, is terminated by the wind reversal at the onset of the austral monsoons. Similarly, for tropical zonal oceanic

variability, maximum interannual western Indian Ocean SST variability is between September and November (Schott and McCreary Jr., 2001), before the MSM. These SST anomalies can persist, and there is a statistically significant relationship between monthly SST and monthly rainfall in northern Madagascar in December (r=0.377, p=0.021), but the relationship does not persist into later monsoon months (Scroxton et al., 2017). At longer timescales MSM rainfall variability appears to respond to both SST variability and meridional atmospheric variability (Scroxton et al., 2017; Scroxton et al., 2019; Voarintsoa et al.,

2019; Zinke et al., 2004) with variability in the spatial teleconnections of El Niño-Southern Oscillation potentially driving subtropical SSTs which influence rainfall (Zinke et al., 2004).

Of relevance to the 4.2kyr event, the response of the MSM to abrupt North-Atlantic cold events appears to be towards wetter conditions, as seen in the response of stalagmite $\delta^{18}$O in northwest Madagascar during the 8.2 kyr event (Voarintsoa et al.,

2019), and in the growth phases of stalagmites in southwest Madagascar during Heinrich stadial 1 and the Younger Dryas (Scroxton et al., 2019). This response fits with the idea of southerly shifts in mean ITCZ position from a cooler Northern





Hemisphere and/or reduced Atlantic thermohaline circulation (Broccoli et al., 2006; McGee et al., 2014; Zhang and Delworth, 2005; Broccoli et al., 2006; McGee et al., 2014; Zhang and Delworth, 2005). A 4.2kyr event forced from a cool North Atlantic (Wang et al., 2013) would therefore predict wet conditions in the MSM and SEAfM more broadly.


[Figure 1 map showing locations in south-west Africa with labels: Kilimanjaro, Lake Challa, Burundi, Lake Masoko, Lake Malawi, Anjohikely/Anjohibe, Rodrigues, Taros Basin, Cold Air Cave, Lake Muzi/Mkhuze Delta]

**Figure 1: Location map of south-west Africa. Blue dots indicate the southern hemisphere summer monsoon regime, defined as where the summer (NDJAM) to winter(MJJAS) rainfall range is greater than 300mm and Monsoon Precipitation Index (summer to winter range/annual precipitation) is greater than 0.5 (Wang and Ding, 2008). Black dots indicate locations of paleoclimate records.**

## 105   3 New samples and methodology

Anjohikely (15.56°S, 46.87°E) is located in the Narinda karst in northwest Madagascar. Sitting in Eocene limestone topped with dolomite, and just 2km SSW of the larger, well-documented Anjohibe, Anjohikely has 2.1km of decorated passages, typically between collapsed dolines but with some well-decorated chambers with more restricted airflow (Laumanns and Gebauer, 1993). From Anjohikely, stalagmite AK1 was extracted in 2014. AK1 is a thin, 830mm tall, candlestick-style,

aragonite stalagmite (Figure 2).

The age model for AK1 was determined from 12 U-Th ages (Table 1). U-Th samples weighing 140–190 mg were prepared and analyzed at the Massachusetts Institute of Technology. Samples were combined with a $^{229}$Th-$^{233}$U-$^{236}$U tracer, digested, purified via iron coprecipitation and ion exchange chromatography. U and Th were analyzed on separate aliquots using a Nu

Plasma II-ES multi-collector ICP-MS equipped with a CETAC Aridus II desolvating nebulizer. U-Th ages were calculated using the half-lives of 75,584 ±110 for $^{230}$Th, 245,620 ±260 for $^{234}$U (Cheng et al., 2013), 1.55125 x 10$^{-10}$ yr$^{-1}$ for $^{238}$U (Jaffey



et al., 1971) and an initial $^{230}$Th/$^{232}$Th ratio of 4.4($\pm$2.2)x10$^{-6}$. Age models was constructed using OxCal (Bronk Ramsey, 2008) using a P-Sequence Poisson process depositional model, with a k0 parameter of 0.1. An additional prior of a hiatus was included at 707mm.


AK1 was sampled for stable isotopes ($\delta^{13}$C and $\delta^{18}$O) at increments ranging from 0.25 to 5mm to achieve an approximately 5-year resolution (min: 13.3 years per sample, max: 0.9, average: 4.5, standard deviation 2.2) (Figure 2). Lower sampling rate sections were drilled, and higher sampling rate sections were milled, both with a 1mm diameter drill bit. A total of 645 samples were analyzed for stable oxygen and carbon isotope ratios using a Thermo Scientific Gas Bench II for sample preparation and
a Thermo Delta V Advantage isotope ratio mass spectrometer at the University of Massachusetts Amherst. Reproducibility of the standards is typically better than 0.04‰ for $\delta^{13}$C and 0.06‰ for $\delta^{18}$O (1$\sigma$).



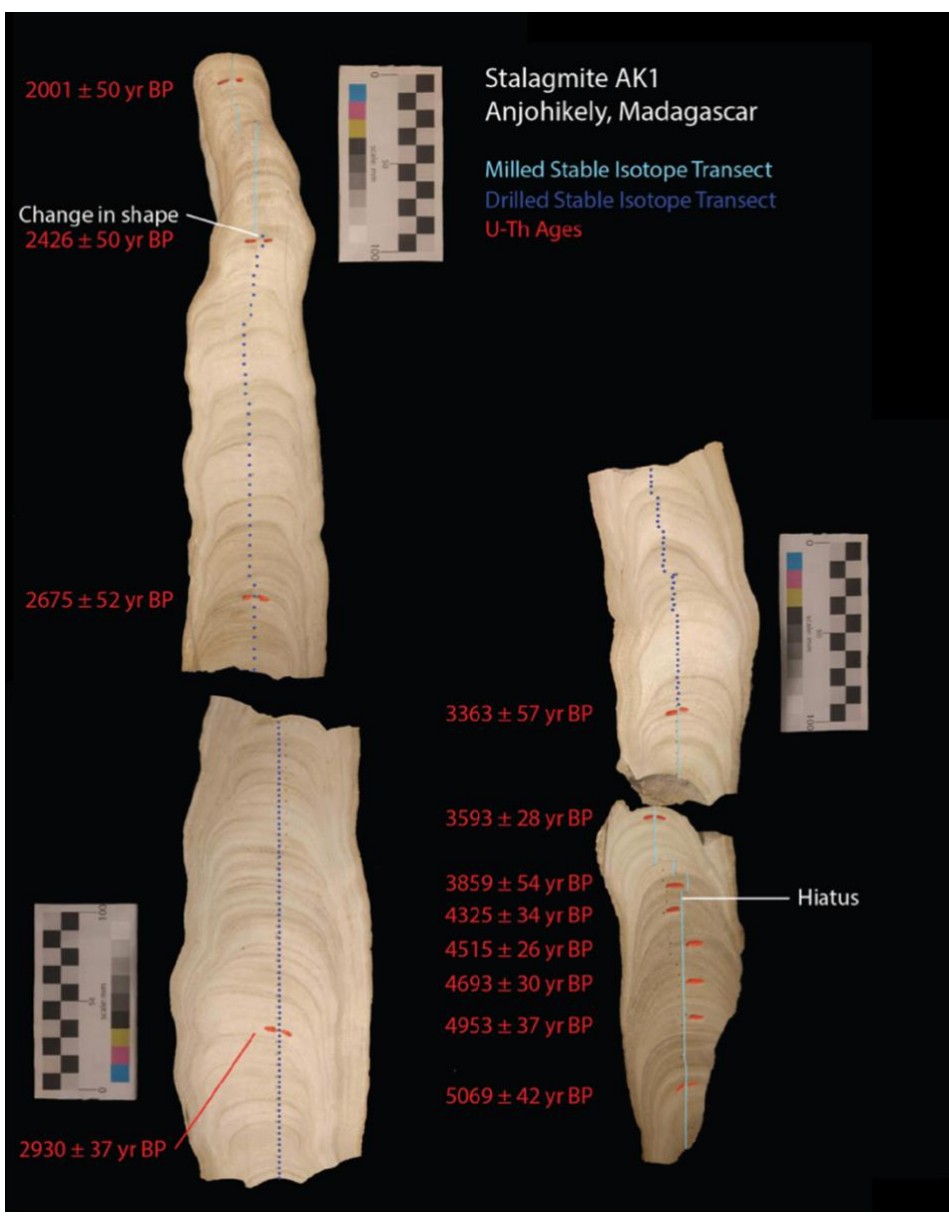

**Figure 2: Photographs of stalagmite AK1 with scalebar length of 100mm. Red shading denotes U-Th sampling locations, dark blue dots show stable isotope drill holes, light blue lines show stable isotope milling trench.**



## 4 Results

### 4.1 AK1 and the local response in Madagascar

Stalagmite AK1 from Anjohikely grew from 5.38 to 1.92 kyr BP with a hiatus between 4.31 and 3.83 kyr BP (Table 1, Figure 3). The $\delta^{18}O$ record is relatively stable before 3.0 kyr BP, with 1 to 2‰ range in centennial scale variability (Figure 2). A

decline in $\delta^{18}O$ between 3.0 and 2.5 kyr BP leads to two significant negative anomalies at 2.65 and 2.4 kyr BP before a return to the least negative $\delta^{18}O$ values at the cessation of growth at 1.9 kyr BP. To the first order we interpret stalagmite $\delta^{18}O$ in northwest Madagascar as a proxy for regional monsoonal strength, likely highly correlated to local rainfall amount, through a combination of the "amount effect" and strength of atmospheric convection (Scroxton et al., 2017; Voarintsoa et al., 2017; Voarintsoa et al., 2019; Wang et al., 2019b). However, the precise mechanisms controlling stalagmite $\delta^{18}O$ response to

hydroclimate changes varies and are discussed in section 5.1.

<Table 1 here, currently at end of manuscript as will need to be horizontal format>

A positive $\delta^{18}O$ excursion at the top of stalagmite AK1 coincides with a change in stalagmite diameter, shape and location of

the drip axis, which are indicative of a change in the drip hydrology or cave ventilation regime. This increases the likelihood of either non-equilibrium deposition and/or enhanced in-karst fractionation. As such, while the positive change in $\delta^{18}O$ is likely indicative of drying conditions, we suggest that the magnitude of $\delta^{18}O$ change in the top 99 mm of AK1 (younger than 2.33 kyr BP) is not directly comparable with the rest of the record.

Between 4.30 and 3.84 kyr there is a growth hiatus, replicated in stalagmite ANJ-94 from Anjohibe at (4.20–3.99) (Wang et al., 2019b). A replicated hiatus likely indicates dry conditions and potentially the driest conditions of the mid/late Holocene. The 4.2 kyr event therefore appears at least locally remarkable in northwest Madagascar. A dry anomaly is the opposite to the wet conditions recorded at 8.2 kyr BP (Voarintsoa et al., 2019), a Holocene climatic anomaly often viewed as a greater magnitude version of the 4.2 kyr event (Bond et al., 2001; Wang et al., 2013).


The largest $\delta^{18}O$ excursions in the AK1 record are two negative (wet) anomalies at 2.65 and 2.40 kyr BP. Both excursions are replicated, within dating errors, as dry events in the Dongge and (to a lesser extent) Sanbao speleothem records from China (Dong et al., 2010; Dykoski et al., 2005) and the Huagapo record from Peru (Kanner et al., 2013). The 2.65 kyr excursion is a dry event in the Sahiya speleothem record of western India (Kathayat et al., 2017). These abrupt hydroclimate anomalies have

received little attention despite being replicable across the tropics and of much greater magnitude there than more frequently studied Holocene climatic events such as the 4.2 kyr event. They are deserving of more thorough investigation in the future.

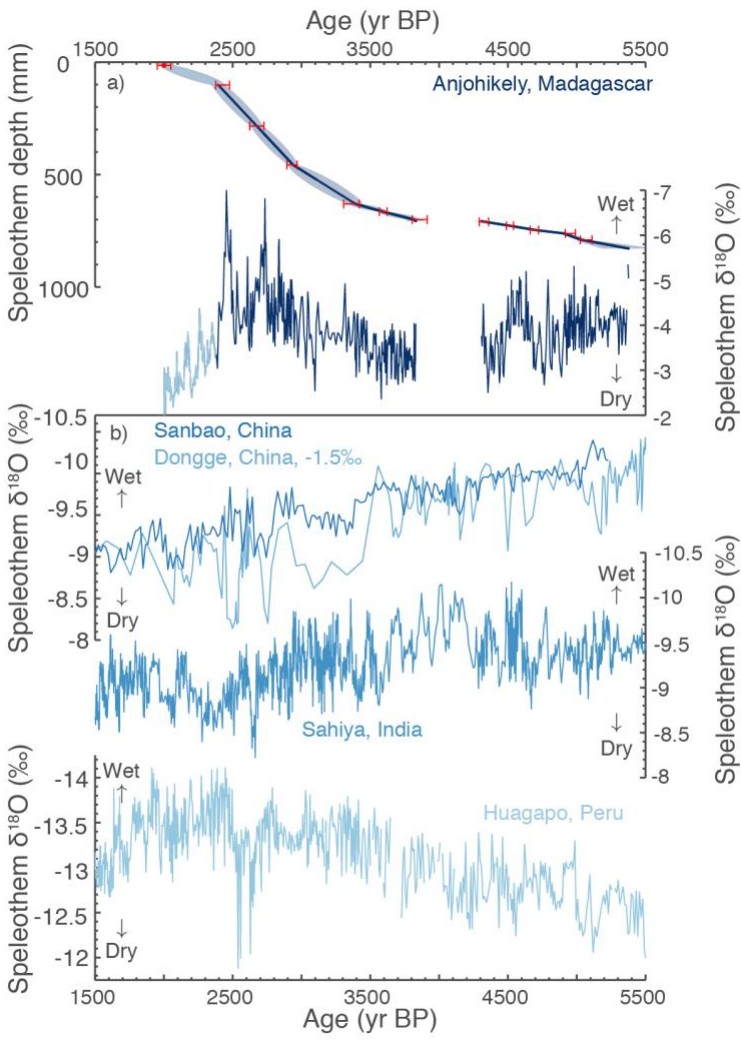

**Figure 3: a) Results from stalagmite AK1 from Anjohikely, northwest Madagascar, showing top: U-Th ages (red error bars), OxCal**
**age model (blue lines) and associated 95% confidence interval (blue shading), and bottom: stalagmite d18O. Data from the top 99mm**
**are shown in a lighter blue. b) comparison with other monsoon influenced speleothem d18O records. From top to bottom: Sanbao**
**and Dongge caves in China, Sahiya cave in India and Huagapo cave in Peru.**

## 4.2 Regional variability in the African monsoons

In the southern hemisphere of East Africa, the Kilimanjaro ice core δ18O shows a gradual drying, accelerating at 3.65 kyr BP

(Thompson et al., 2002)(Figure 4). An increase in dust occurs at 4.2 kyr BP but the isotopes indicate only a gradual change

from warmer and wetter conditions to dry and cooler. A pollen-based estimate of precipitation from multiple sites in Burundi

suggest a transition from relatively stable conditions to higher-amplitude swings between low and high precipitation around



3.6 kyr BP, but no abrupt 4.2 kyr event (Bonnefille and Chalie, 2000; Bonnefille and Chalie, 2000) . It is questionable whether

the Lake Challa leaf wax δD and BIT index have the resolution to record an abrupt 4.2 kyr event. The low-resolution BIT record shows a peak in wet conditions between 4.2 and 3.7 kyr BP, but this is part of a long-term millennial scale trend lasting 1.5 kyr (Verschuren et al., 2009). The leaf wax δD is inverse, indicating peak dry conditions between 4.2 and 3.7 kyr BP, again part of a longer millennial scale trend. The authors reconcile these differences suggesting that the δD likely records moisture transport processes than local rainfall amount (Tierney et al., 2011; Tierney et al., 2011) , and attribute changes to tropical

zonal reorganisation. In all cases observed change fit with tropical reorganisation at 4.0 kyr BP better than an abrupt, 300 year long 4.2 kyr event. We interpret all four East African monsoon records as showing no sign of an abrupt 4.2 kyr event.

Further south in the SEAfM, at Lake Masoko drying begins around 4.6 kyr BP, peaking around 4.3 kyr BP (Garcin et al., 2006). A possible short hiatus occurs between 4.0 and 3.9 kyr BP. At Lake Malawi drying begins around 4.65 kyr BP. Between

4.4 and 3.95 kyr BP there is only a single datapoint, which given surrounding deposition rates, we interpret as an interpolated point through an unrecognised hiatus (Johnson et al., 2002).

In the Indian Ocean pollen counts (ln(*Latania/Eugenia*)) from the Taros Basin in Mauritius suggest dryer conditions between 4.5 and 4.1 kyr BP, while sediment core ln(Ca/Ti) ratios indicate brief centennial wet events at 4.38 and 4.15 kyr BP, all on a

background shift from wetter to dryer conditions at 4.8 kyr BP (de Boer et al., 2014). Further east on Rodrigues, speleothem δ18O values from La Vierge show no change in conditions at the 4.2 kyr event but do show a gradual drying beginning around 3.9 kyr, interpreted as part of the widespread tropical climatic changes at this time (Li et al., 2018).

In South Africa, at Cold Air Cave there is little change in speleothem δ18O over the 4.2 kyr event (Holmgren et al., 2003), with

slightly wetter conditions between 4.6 and 4.05 kyr BP and slightly dry conditions between 4.05 to 3.8 kyr BP. A growth phase of stalagmite T5 between 4.35 and 3.95 kyr BP suggests wetter conditions during the Middle to Late Holocene transition but could be a coincident change in drip hydrology (Repinski et al., 1999). Sediment cores from Lake Muzi (Humphries et al., 2019) and Mkhuze Delta (Humphries et al., 2020) in eastern South Africa both indicate periods of wet conditions between 4.25 and 3.8 kyr BP.





**Figure 4: Regional hydroclimate changes in southeast Africa between 5 and 3 kyr BP. For each record proxy z-score is calculated \
between 2.5 and 5.5 kyr BP to reduce the influence of orbital scale changes. Circles indicate datapoints. Lines without circles are at**



**higher resolution so circles have been omitted for clarity. Blue bars indicate the duration of the 4.2kyr event and a regional hydroclimatic change at 4.6 kyr BP. Records are plotted so that wet conditions are up.**

## 5 Discussion

### 5.1 Replication of stalagmites of mid-late Holocene climate in northwest Madagascar

Replication of results from the same or nearby caves is considered the gold standard for producing reliable climate records from stalagmite proxy time series (Dorale and Liu, 2009). Two speleothem $\delta^{18}O$ records from northwest Madagascar record the 5000 to 3000-year BP interval: AK1 from Anjohikely (this study) and ANJ94-5 from Anjohibe (Wang et al., 2019b). Anjohibe is 2.3km northeast of Anjohikely. Both have hiatuses at the 4.2 kyr event: 4.3–3.8 in AK1, 4.2–4.0 in ANJ94-5. ANJ94-5 shows a slightly later cessation of growth and a positive excursion into the event, potentially due to progressive enrichment of a dwindling karst water store. AK1 also shows minor $\delta^{18}O$ enrichment (0.7‰ over 1.5mm or 13 years) just before the hiatus. The positive excursions seen in both stalagmites leading into the hiatus is evidence that the hiatus was caused by dry rather than wet conditions. Therefore, the primary result of this paper is replicated.

However, the $\delta^{18}O$ records of speleothems ANJ94-5 and AK1 do not overlie each other and do not initially appear to replicate. Here we discuss where $\delta^{18}O$ records disagree, where $\delta^{18}O$ records might agree with other hydroclimate indicators such as growth rate, and what might be the possible causes. ANJ94-5 is a mixed mineralogy stalagmite, whereas AK1 is aragonitic. The aragonitic sections of ANJ94-5 at 4.8–4.6 kyr BP and 4.0kyr BP onwards have $\delta^{18}O$ values comparable to those of AK1. However, the isotopic difference between calcite and aragonitic sections of ANJ94-5 of ~2‰ is far larger than the expected offset between calcite and aragonite of ~0.8‰ determined from laboratory studies (Kim et al., 2007), theoretical calculations (Tarutani et al., 1969), and in stalagmites from Anjohibe (Scroxton et al., 2017).

The discrepancies between $\delta^{18}O$ records could be explained by differences in cave conditions. ANJ94-5 was collected from a chamber open to the atmosphere, with atmospheric $CO_2$ concentrations (Wang et al., 2019b). ANJ94-5 was therefore likely subject to considerable kinetic fractionation during speleothem growth (Mickler et al., 2006). Anjohikely has more restricted chambers and a greater areal coverage of precipitated calcite, especially on the walls and floor. Therefore, while additional evidence from ANJ94-5 suggests that isotopic variability may still be climatic in origin (Wang et al., 2019a), the absolute $\delta^{18}O$ values are likely not comparable with AK1, sourced from a more restricted chamber in a 'wetter' cave.

With this in mind, a comparison of more positive and negative periods of $\delta^{18}O$ in both stalagmites does show good reproducibility interpreted as broad-scale climatic changes in the hydrological cycle. Both stalagmites show a gradual positive

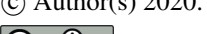



(drying) trend between 5500 and 4200 kyr BP with modest centennial scale variability indicated by more negative $\delta^{18}$O (wetter)(5.3–5.2,5.1–4.9, around 4.5 kyr BP) and more positive $\delta^{18}$O (drier)(4.9–4.7, 4.5–4.3 kyr BP) values.


After (above) the hiatus there is agreement between the growth rate of ANJ94-5 and the isotopes of AK1. Between 3.15 and 2.4 kyr BP there is a 0.7‰ $\delta^{18}$O decrease in AK1, suggesting wetter conditions (3.5 to 3.15 kyr BP: -3.7‰, 3.15 to 2.4 kyr BP: -4.4‰). ANJ94-5 also has a negative $\delta^{18}$O excursion, but it is smaller at around 0.3‰ (3.5 to 3.15 kyr BP: -4.0 ‰, 3.15 to 2.4 kyr BP: -4.3‰). An increased growth rate in ANJ94-5 is also likely indicative of wetter conditions, either through greater

transport of calcium ions by reduced PCP or enhanced flow rate, or enhanced vegetative activity increasing soil $p$CO$_2$ and the dissolution of the karst host rock.

Both stalagmites return to higher $\delta^{18}$O (drier conditions) at 2.4kyr BP. AK1: 3.15 to 2.4 kyr BP: -4.4‰, 2.4 to 2.0 kyr BP: -3.0‰. ANJ94-5: 3.15 to 2.4 kyr BP: -4.3‰, 2.4 to 2.0 kyr BP: -3.9‰. However, at this point AK1 undergoes a shape change,

becoming thinner and less cylindrical. We suggest that from 2.4 kyr BP onwards, AK1 may also be subject to enhanced disequilibrium effects, perhaps related to changes in cave ventilation regime, and/or a progressive drying of the drip prior to the termination of growth. We suggest that the isotopic values during this section (younger than 2.4 kyr BP) are not directly comparable to those elsewhere in the stalagmite.

In addition to isotopic differences caused by kinetic fractionation, it is also possible that different drip pathways contribute to different isotopic responses in the two stalagmites. For example, differences in storage and mixing and in-karst evaporation during the dry season (Markowska et al., 2020) might lead to different sensitivities to different parts of the hydrologic system: extreme events, seasonal vs long-term mean etc.

The consequences of different cave conditions, karst storage and drip pathways on stalagmite $\delta^{18}$O remains a working hypothesis. More efforts are needed focusing on replicating northwest Madagascar speleothem $\delta^{18}$O and understanding the local hydrology at a drip rather than cave level.





**Figure 5: Comparison of speleothems from Anjohikely (AK1, blue colors, this study) and Anjohibe (ANJ94-5, red colors 36), two**
**caves less than 2 km apart in northwest Madagascar. a) speleothem δ¹⁸O during the period of overlap. b-e) 1000-year close-up of events around the 4.2 kyr BP event indicating the contemporaneous hiatus in both speleothems. b,d) Age depth model, circles indicate individual stable isotope data points linked by line. Shading denotes 2σ age model error for stalagmite AK1. Light colored error bars show individual dates with 2s error. c,e) individual δ¹⁸O measurements for each stalagmite.**





## 5.2 Middle to Late Holocene hydroclimate changes in the southeast African monsoon?

A hydroclimate event synchronous to the 4.2 kyr event appears to have some local significance in the SEAfM domain, particularly around northern Madagascar and Lakes Malawi and Masoko. Peak dry conditions occur 4.5–4.1 kyr BP at Lake Masoko, 4.4–4.0 kyr BP at Lake Malawi, 4.2–4.0 kyr BP at Anjohibe, 4.3–3.8 kyr BP at Anjohikely, and 4.5–4.1 kyr BP at Taros Basin. Peak wet conditions occur at 4.4–3.95 at Cold Air cave (based on stalagmite T5), 4.2–3.8 kyr BP at Lake Muzi and 4.2–3.9 kyr BP at Mkuze Delta. The age errors for most records are around ±600 years (2σ) for the stalagmite records and

±200 years (2σ) for most other records. Therefore, these hydroclimate anomalies are all potentially synchronous with the 4.2 kyr event (4.26–3.97 kyr BP).

The spatial pattern of hydroclimate anomalies at the 4.2 kyr event approximates the spatial pattern of hydroclimate anomalies during weak MCT years. In the modern climate, weak MCT years (1981, 1990, 2006, 2017)(Figure 6a) result in dry conditions

in northern Mozambique, Madagascar and Mauritius, wet conditions over South Africa, weakly dry conditions over Malawi and weakly wet conditions over Burundi, and Tanzania (Barimalala et al., 2020; Xie and Arkin, 1997). This suggests that the 4.2 kyr event may be locally expressed as a period of more frequent weak Mozambique Channel Trough events. Further comparison with ERA-Interim reanalysis of the 850hPa specific humidity (Figure 6b) shows a similar pattern, indicating the rainfall anomalies are associated with decreased moisture convergence over the northern Mozambique channel (Barimalala et

al., 2020; Dee et al., 2011). Some mismatches occur at Rodrigues and possibly Lake Muzi and Mkhuze Delta. SST anomalies suggest decreased rainfall is associated with higher subtropical SSTs to the south-east of Madagascar, and cooler tropical SSTs to the north-east of Madagascar (Barimalala et al., 2020). Local SSTs in the source regions of the northern Mozambique Channel and equatorial West Indian oceans show a slight but non-significant cooling.

We suggest the 4.2 kyr event is associated with a period of more frequent weak Mozambique Channel Trough events, where reduced cyclonic conditions, atmospheric convergence and recurving of moisture bearing winds over the Mozambique Channel and onto Madagascar leads to reduced rainfall in the MSM. Weak MCT years are associated with a northerly location of the summer ITCZ relative to its climatological mean in the west Indian Ocean (Barimalala et al., 2020; Barimalala et al., 2020). Therefore, we hypothesize that the southern hemisphere summer ITCZ over the western Indian Ocean was further north during

the 4.2 kyr event.





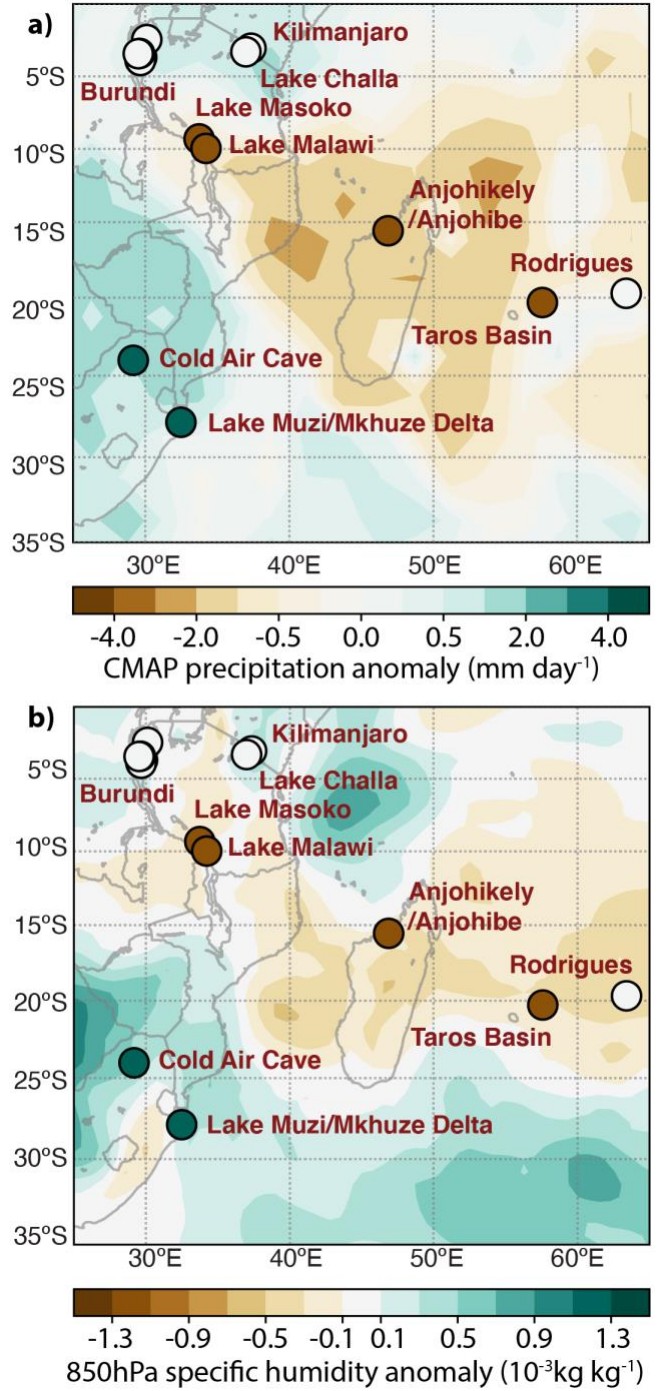

**Figure 6: Combined modern and paleo- climate anomaly maps. Coloured circles indicate wet (green), dry (brown) or no (white) anomaly during the 4.2 kyr event at individual sites. Map colours indicate a) CMAP precipitation anomaly (Xie and Arkin, 1997; Xie and Arkin, 1997) and b) ERA-Interim (1980-2017) reanalysis specific humidity anomaly at 850hPa (Dee et al., 2011) for weak Mozambique Channel Trough years: 1981, 1990, 2006, 2017. Figures based on (Barimalala et al., 2020).**



### 5.3 Timing of the middle to late Holocene climate shifts in the SEAfM

While peak anomalies overlap within age uncertainty of the 4.2 kyr event, a causal relationship should not automatically be inferred. In most of the paleoclimate records in this compilation the Middle to Late Holocene hydroclimate anomaly begins

around 4.6 kyr BP and is frequently gradual. This is earlier than, and in contrast to the abrupt 4.26 kyr BP onset of the 4.2 kyr event in the Mediterranean and Middle East. Drying begins around 4.6 kyr BP at Lake Masoko, Lake Malawi, and the Taros Basin. Wet conditions begin around 4.6 kyr BP at the Mkhuze Delta. At Anjohibe, Anjohikely and Lake Muzi, the three records with hydroclimatic changes very close to 4.26 kyr BP, there is also a hydroclimate change at 4.6 kyr BP, but these changes exhibit the opposite sign to the 4.2 kyr BP changes. Therefore, it should not yet be concluded that these hydroclimate anomalies

are part of the 4.2 kyr event without further evidence as to the climatic mechanisms behind the event, which are currently lacking. Further, even if the 4.2 kyr BP event is present, other hydrological changes during the same millennium (notably around 4.6 kyr BP) seem to have similar or even stronger regional coherence.

Finally, it is worth noting the general lack of expression of the 4.0 kyr tropical climate shift in the MSM, especially when

compared with the more equatorial East African Monsoon records that indicate a reorganisation of tropical zonal climate (Gagan et al., 2004; Marchant and Hooghiemstra, 2004). Modern tropical zonal variability also has reduced influence on the MSM due to its atmospheric isolation from the impacts of the Indian Ocean Dipole by the seasonally locked IOD atmospheric anomalies (Schott and McCreary Jr., 2001). The MSM is also likely responsive to changing sea-surface temperatures (Koseki and Bhatt, 2018; Scroxton et al., 2019). Together, this would suggest that the 4.0 kyr BP tropical climate shift was not

associated with changing west Indian Ocean sea-surface temperatures, but rather was forced by a change in eastern Indian Ocean or Pacific SSTs, leading to an atmosphere only response in the western Indian Ocean and limited impact on rainfall amount in the MSM. High resolution eastern Indian Ocean SST records are not yet available to test this hypothesis.

### 6 Conclusions

Stalagmites from Anjohibe (Wang et al., 2019b; Wang et al., 2019b) and Anjohikely (this study) caves show replicated hiatuses

beginning near 4.2kyr BP, indicating likely dry conditions in northwest Madagascar. Alongside dry conditions at Lake Masoko and Lake Malawi, this observation provides evidence for a locally significant hydroclimate anomaly coincident with the 4.2 kyr event. The response on Madagascar is opposite to the local response to 8.2 kyr event (Voarintsoa et al., 2019) indicating a fundamentally different climate mechanism. The spatial pattern of peak hydroclimate anomalies around 4.2 kyr BP matches the conditions seen in years with a weak Mozambique Channel Trough (MCT), suggesting the 4.2 kyr event may have been a

time with more frequent weak MCT occurrences. Weak MCT years are associated with a northerly position of the summer west Indian Ocean ITCZ.

However, many regional hydroclimate anomalies fail to provide evidence of an abrupt 4.2 kyr event. Hydroclimate changes in the middle to late Holocene are typically gradual and begin earlier than the abrupt 4.2 kyr event, casting doubt as to whether
the 4.2 kyr event could be the cause of regional hydroclimate anomalies at this time. Assuming causality of the entire regionally coherent hydroclimate anomaly pattern would be an overinterpretation without further understanding of the mechanistic processes behind the 4.2 kyr event.

## Data availability

Data are available from the authors (nick.scroxton@ucd.ie), at the NOAA Paleoclimatology Database:
https://www.ncdc.noaa.gov/paleo/study/xxxxx, and have been submitted to the SISAL database.

## Author Contributions

NS ran stable isotope and U-Th chemistry analysis of stalagmite AK1 in the labs of SJB and DM. NS conducted data analysis and was primarily responsible for writing the manuscript. SJB and DM conducted preliminary laboratory analysis and helped write and edit the manuscript. LRG contributed to the manuscript at all stages. PF conducted U-Th chemistry. SJB, LRG, LR
and PF conducted the fieldwork and speleothem collection.

## Competing interests

The authors declare no competing interests.

## Acknowledgements

NS, SJB and DM acknowledge support from NSF award AGS-1702891/1702691, LRG and SJB from NSF award BCS-
1750598 and DM from NSF award EAR-1439559 and the MIT Ferry Fund. Fieldwork in northwest Madagascar was conducted under a collaborative accord for paleobiological research between the University of Antananarivo (Département de Paléontologie et d'Anthropologie Biologique) and the University of Massachusetts (Department of Anthropology); collaborative work was further supported under a second accord for paleobiological and paleoclimatological research between the University of Antananarivo (Mention Bassins sédimentaires, Evolution, Conservation) and the University of Massachusetts
Amherst (Departments of Anthropology and Geosciences). The research was sanctioned by the Madagascar Ministry of Mines, the Ministry of Education, and the Ministry of Arts and Culture.



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





| Sample ID | Depth (mm) | 238U (ng/g)[a] | ±(2σ) | 232Th (pg/g)[a] | ±(2σ) | δ234U (per mil)[b] | ±(2σ) | (230Th/238U) activity | ±(2σ) | 230Th/232Th ppm atomic | ±(2σ) | Age (yr) (uncorr)[c] | ±(2σ) | Age (yr) (corr)[d] | ±(2σ) | δ234U initial (per mil)[e] | ±(2σ) | Age (yr BP) (corr)[f] | ±(2σ) |
|---|---|---|---|---|---|---|---|---|---|---|---|---|---|---|---|---|---|---|---|
| AK1-15 | 14.5 | 5200 | 100 | 2708 | 57 | -3.5 | 1.5 | 0.01883 | 0.00045 | 574 | 14 | 2081 | 50 | 2066.4 | 9.5 | -3.6 | 1.5 | 2001 | 50 |
| AK1-99 | 102. | 5070 | 100 | 1452 | 34 | -2.4 | 1.5 | 0.02260 | 0.00044 | 1254 | 29 | 2499 | 50 | 2491.4 | 8.4 | -2.4 | 1.5 | 2426 | 50 |
| AK1-275 | 284. | 4886 | 98 | 922 | 26 | -3.3 | 1.4 | 0.02477 | 0.00046 | 2083 | 56 | 2745 | 52 | 2739.5 | 8.2 | -3.3 | 1.4 | 2675 | 52 |
| AK1-450 | 457.5 | 6480 | 130 | 272 | 17 | -1.3 | 1.3 | 0.02706 | 0.00033 | 10230 | 640 | 2996 | 37 | 2994.6 | 8.1 | -1.3 | 1.3 | 2930 | 37 |
| AK1-625 | 631. | 4579 | 92 | 287 | 19 | -1.1 | 1.3 | 0.03093 | 0.00050 | 7840 | 510 | 3429 | 57 | 3427.8 | 9.1 | -1.1 | 1.3 | 3363 | 57 |
| AK1-665 | 665. | 3201 | 64 | 197 | 16 | -2.0 | 2.2 | 0.03298 | 0.00023 | 8510 | 680 | 3664 | 28 | 3662. | 28. | -2.0 | 2.3 | 3593 | 28 |
| AK1-693 | 700. | 2980 | 60 | 1804 | 49 | -4.2 | 2.1 | 0.03539 | 0.00046 | 928 | 21 | 3946 | 53 | 3928. | 54. | -4.3 | 2.1 | 3859 | 54 |
| AK1-710 | 710.5 | 2950 | 59 | 364 | 19 | -3.3 | 2.2 | 0.03940 | 0.00028 | 5080 | 240 | 4398 | 34 | 4395. | 34. | -3.4 | 2.2 | 4325 | 34 |
| AK1-727 | 727.5 | 4126 | 83 | 563 | 20 | -2.6 | 1.9 | 0.04110 | 0.00021 | 4790 | 140 | 4589 | 25 | 4585. | 25. | -2.6 | 1.9 | 4515 | 26 |
| AK1-746 | 746.5 | 3559 | 71 | 477 | 18 | -0.8 | 2.0 | 0.04274 | 0.00025 | 5060 | 170 | 4766 | 30 | 4762. | 30. | -0.8 | 2.0 | 4693 | 30 |
| AK1-761 | 761.5 | 4093 | 82 | 2191 | 47 | -5.2 | 1.9 | 0.04492 | 0.00030 | 1332 | 13 | 5039 | 36 | 5022. | 37. | -5.2 | 1.9 | 4953 | 37 |
| AK1-763 | 790. | 7130 | 140 | 1076 | 28 | 0.5 | 1.2 | 0.04605 | 0.00037 | 4849 | 92 | 5138 | 42 | 5134. | 13. | 0.5 | 1.2 | 5069 | 42 |





**Table 1: U-Th dating table for stalagmite AK1**

**a Reported errors for $^{238}$U and $^{232}$Th concentrations are estimated to be ±1% due to uncertainties in spike concentration; analytical uncertainties are smaller.**

**b $\delta^{234}$U = ([$^{234}$U/$^{238}$U]$_{activity}$ - 1) x 1000.**

**c [$^{230}$Th/$^{238}$U]$_{activity}$ = 1 - e$^{-\lambda 230 T}$ + ($\delta^{234}$U$_{measured}$/1000)[ $\lambda_{230}$/( $\lambda_{230}$ - $\lambda_{234}$)](1 - e$^{-(\lambda 230 - \lambda 234)\,T}$), where T is the age. "Uncorrected" indicates that no correction has been made for initial $^{230}$Th.**

**d Ages are corrected for detrital $^{230}$Th assuming an initial $^{230}$Th/$^{232}$Th of (4.4±2.2) x 10$^{-6}$.**

**e $\delta^{234}$U$_{initial}$ corrected was calculated based on $^{230}$Th age (T), i.e., $\delta^{234}$U$_{initial}$ = $\delta^{234}$U$_{measured}$ X e$^{\lambda 234 * T}$, and T is corrected age.**

**f B.P. stands for "Before Present" where the "Present" is defined as the January 1, 1950 C.E.**