# Peer review of "Possible expression of the 4.2 kyr event in Madagascar and the southeast African monsoon"

_Climate of the Past, 2020_

## Short Comment (SC1) · 28 Dec 2020

See attached pdf with the figure

Comment for: Scroxton, N., Burns, S. J., McGee, D., Godfrey, L. R., Ranivo-harimanana, L., and Faina, P.: Possible expression of the 4.2  kyr event in Madagascar and the south-east African monsoon, Clim. Past Discuss. [preprint], https://doi.org/10.5194/cp-2020-137, in review, 2020.

This manuscript is interesting but, in my opinion, is lacking some fundamental aspects to fully comprehend the stable oxygen isotope proxies that the author used to reconstruct paleoclimate in Anjohikely. Among these are the description of the cave and its microclimate, where in the cave was it collected, and importantly what are the potential

factors that drive oxygen isotope variability inside that cave.

About the cave:

In an article published in 1997, Burney and colleagues described (p. 756) that Anjohikely is linked to Anjohibe with a subterranean passage (while citing Laumans et al., 1991). In addition to this, Burney et al. (1997) noted that "most passages in Anjohikely are relatively small in diameter, with somewhat limited development of speleothems (stalactites, stalagmites, and other dripstone formations). This makes me wonder about this sample from Anjohikely, particularly that its size is relatively big if considering the former description and speleothem investigation by Burney et al.

It may be helpful if the authors provide, at least, a skecth or map of the cave, with some illustrative figures of the cave entrance and the chamber from where the speleothems were extracted.

It would also be helpful to have a brief note about the overall microclimatic condition inside the cave (what is the air temperature inside, how about pCO2 and relative humidity [RH])? Somewhere in the discussion (L. 224-229) that the authors discuss about kinetic fractionation as processes affecting stalagmites collected from Anjohibe (Wang et al., 20190) versus stalagmites from Anjohikely (their work). If the cave atmospheric exchange with its exterior atmosphere sounds important, information about cave microclimate (T, pCO2, RH) could be crucial and discussed with more details. Also, I am far from believing that stalagmite growing in Anjohikely was precipitating in equilibrium with the cave drip water, as several studies have proven that almost none of the terrestrial carbonates precipitate in isotopic equilibrium with their drip water (Mickler et al., 2006; Tremaine et al., 2011; Day and Henderson, 2011; Deininger et al., 2012, Daëron et al., 2019).

About some statements in the paper: I disagree with the statement at Line 44 that "the impact of the 4.2 kyr event on the tropics and subtropics is unknown". • Please consult Railsback et al., 2019, QSR, in which a review of the 4.2 ka event was presented with a new isotopic record from Namibia.   As the authors also make a comparison with another stalagmite from Anjohibe about the 4.2 ka (Wang et al., 2019 QSR), this statement "unknown" needs to be revised.   Other relevant information from India could be known by reading Kathayat et al. (2018, CP) and from China (Zhang et al., 2019 CP)   Some comments in this section may be applicable to the preprint of the same authors (Scroxton, N., Burns, S. J., McGee, D., Godfrey, L. R., Ranivoharimanana, L., and Faina, P.: Circum-Indian ocean hydroclimate at the mid to late Holocene transition: The Double Drought hypothesis and consequences for the Harappan, Clim. Past Discuss. [preprint], https://doi.org/10.5194/cp-2020-138, in review, 2020) I am also curious to know the opinion of the authors about the two replicated mid-Holocene hiatuses reported in Voarintsoa et al. (2017 CP) in this region (one in Anjohibe, and another in Anjokipoty). As Anjohikely is a small cave, it is expected to behave like Anjokipoty (hence, should record the same hiatus of mid-Holocene deposition), and this comes back to my point earlier about the need of more details about the cave.

Interpretation of the data In Figure 2: the authors indicate a hiatus within the palest color of the oldest generation of stalagmite. I wonder if this hiatus should actually be located at the boundary between the pale-brown color and the whiter color stalagmite (see annotated figure below), and if each of the color change throughout the sample too can indicate other short term growth hiatus? As a matter of fact, only the bottom stalagmite has more age data. In contrast, the upper part of the stalagmite has lesser trenches. Why is that? Has there be any diagenesis at that brown-pale bottom part that allowed loss of U, hence the samples appear older?

At L. 150: Growth hiatus, what are/is the rational for saying "there is a growth hiatus"? I wonder what are the rational for saying that it indicates dry or wet conditions? While looking at the time series in Figure 3, it appears that the isotopic value bracketing the so-called hiatus are showing more negative values. Wouldn't this hiatus represent a wet condition? (May be an evaluation of the petrography (e.g., Railsback et al., 2013)

would be useful here.

Presentation of the manuscript:

I feel that the authors should clearly write a results section and not combine results with discussion, or vice versa. Some short interpretations of the results are acceptable, if these are meant to emphasize the findings, but results should report results. With that said,   In section 4.1 they should elaborate on the isotopic range, if there are any periodicity, highlight the extreme positive/negative excursion, and provide evidence of the hiatus. The author should also discuss about the growth rate. By looking at their Figure 3a, it seems that growth rate of the bottom part of the sample is slow vs. the upper part of the sample.   I also feel that some information presented in the discussion belong to the results section (if not mentioning paragraph at L 230, L 235, and L. 242)

The section about the regional variability in the African monsoon (Section 4.2) does not seem to belong into the results section.

Other detailed comments: At L 145: Can you please elaborate, or be specific, on the statement "change in drip hydrology" and "change in cave ventilation regime"?

Figure 3: Can you please replace "Speleothem depth" with "distance from the top of the speleothem? In my understanding, depth is most commonly applied to sediments that are dig underground.

The authors mention in passing the diameter, the shape, and location of the drip axis (e.g., L. 245), it would be better to apply the layer-bounding surfaces approach (such approach was used in Wang et al., 2019 QSR) to quantify such changes. About the shape of the stalagmite again, I think there is quite a number of literature that they could use to back up their statement.

Minor editorial errors: For some reasons, several of the in-text citations are replicated, if not only mentioning some at Lines 31, 36, 40, and 73). I guess some attention from

the authors to avoid such replication is appreciated

If you use aragonitic, then calcitic seems to be parallel. However, it may be better to use aragonite and calcite (e.g., aragonite section..)

References:

Burney, D.A., James, H.F., Grady, F.V., Rafamantanantsoa, J.G., Ramilisonina, Wright, H.T., Cowart, J.B., 1997. Environmental change, extinction and human activity: evidence from caves in NW Madagascar. Journal of Biogeography 24, 755-767.

Kathayat, G., Cheng, H., Sinha, A., Berkelhammer, M., Zhang, H., Duan, P., Li, H., Li, X., Ning, Y., and Edwards, R. L.: Evaluating the timing and structure of the 4.2 ka event in the Indian summer monsoon domain from an annually resolved speleothem record from Northeast India, Clim. Past, 14, 1869–1879, https://doi.org/10.5194/cp-14-1869-2018, 2018.

Railsback, L.B., Liang, F., Brook, G.A., Voarintsoa, N.R.G., Sletten, H.R., Marais, E., Hardt, B., Cheng, H., Edwards, R.L., 2018. The timing, two-pulsed nature, and variable climatic expression of the 4.2 ka event: A review and new high-resolution stalagmite data from Namibia. Quaternary Sci Rev 186, 78-90.

Railsback, L.B., Akers, P.D., Wang, L., Holdridge, G.A., Voarintsoa, N.R.G., 2013. Layer-bounding surfaces in stalagmites as keys to better paleoclimatological histories and chronologies. International Journal of Speleology 42, 167-180.

Daëron M., Drysdale R. N., Peral M., Huyghe D., Blamart D., Coplen T. B., Lartaud F. and Zanchetta G. (2019) Most Earth-surface calcites precipitate out of isotopic equilibrium. Nat Comm. 10, 1–7. Day C. and Henderson G. (2011) Oxygen isotopes in calcite grown under cave-analogue conditions. Geochim Cosmochim Ac 75, 3956–3972. Deininger M., Fohlmeister J., Scholz D. and Mangini A. (2012) Isotope disequilibrium effects: The influence of evaporation and ventilation effects on the carbon and oxygen isotope composition of speleothems–A model approach. Geochim

Cosmochim Ac 96, 57–79. Mickler P. J., Stern L. A. and Banner J. L. (2006) Large kinetic isotope effects in modern speleothems. GSA Bulletin 118, 65–81. Tremaine D. M., Froelich P. N. and Wang Y. (2011) Speleothem calcite farmed in situ: Modern calibration of $\delta$18O and $\delta$13C paleoclimate proxies in a continuously-monitored natural cave system. Geochim Cosmochim Ac 75, 4929–4950. Zhang, H., Cheng, H., Cai, Y., Spötl, C., Kathayat, G., Sinha, A., Edwards, R. L., and Tan, L.: Hydroclimatic variations in southeastern China during the 4.2 ka event reflected by stalagmite records, Clim. Past, 14, 1805–1817, https://doi.org/10.5194/cp-14-1805-2018, 2018. Voarintsoa, N. R. G., Railsback, L. B., Brook, G. A., Wang, L., Kathayat, G., Cheng, H., Li, X., Edwards, R. L., Rakotondrazafy, A. F. M., and Madison Razanatseheno, M. O.: Three distinct Holocene intervals of stalagmite deposition and nondeposition revealed in NW Madagascar, and their paleoclimate implications, Clim. Past, 13, 1771–1790, https://doi.org/10.5194/cp-13-1771-2017, 2017.

Please also note the supplement to this comment:
https://cp.copernicus.org/preprints/cp-2020-137/cp-2020-137-SC1-supplement.pdf

---

## Referee Comment (RC1) · Anonymous Referee #1 · 30 Dec 2020

Climate of the Past Submission cp-2020-137, "Possible expression of the 4.2 kyr event in Madagascar and the southeast African monsoon" by Scroxton et al., presents evidence from stalagmite AK1 from Anjohikely cave, northwest Madagascar and makes inferences about climate from 5000 to 2000 years BP. The most profound inference is of a "period of drought that lasted continuously from $\sim$ 4.32 and 3.83 ka BP". The authors have highlighted the hiatus recorded in the stalagmite AK1 between 4.32 and 3.83 kyr BP, replicating a hiatus in another stalagmite from nearby Anjohibe, and therefore indicating a significant drought around the time of the 4.2 kyr event in the region. The fundamentals of this research project are entirely based on the hiatus recorded in the stalagmites.

The study draws on a stalagmite, AK-1, from Anjohikely cave, northwest Madagascar.

[Figure]

I infer that the manuscript draws its scientific conclusions about the 4.2. ka event is from the hiatus recorded in the stalagmite AK-1, and comparisons with the previously published studies.

Suggestion 1. I think it should be, hence, provide the explicit petrographic studies of the hiatus. The layer bounding studies as discussed by Railsback et al., 2013 is important for this project because it will provide robust evidence demonstrating if the periods of non-deposition, either because of exceptionally wet or dry conditions.

Reference : Railsback, L.B., Akers, P.D., Wang, L., Holdridge, G.A., Voarintsoa, N., 2013. Layer-bounding surfaces in stalagmites as keys to better paleoclimatological histories and chronologies. Int. J. Speleol. 42, 167–180.

―――――――――――――――――――――――

---

## Editor Comment (EC1) · Denis-Didier Rousseau (Editor) · 31 Dec 2020

Dear Authors,

As reviewer 1 released some comments, could you please post a short reply?

All the very best

denis-didier Rousseau

---

## Author Comment (AC1) · 5 Jan 2021

"This manuscript is interesting but, in my opinion, is lacking some fundamental aspects to fully comprehend the stable oxygen isotope proxies that the author used to reconstruct paleoclimate in Anjohikely. Among these are the description of the cave and its microclimate, where in the cave was it collected, and importantly what are the potential factors that drive oxygen isotope variability inside that cave."

We agree with this statement in general terms. The recent increase over the last five years on stalagmites from the area have now produced d18O records covering the vast majority of the Holocene. None of those studies have yet laid out a detailed explanation on the controls of d18O in the region.

[Figure]

Given the highly seasonal nature of rainfall at the site, the single source of precipitation, and its proximity to the ocean it is highly likely that the amount effect dominates. However other factors such as evaporative enrichment in the karst (eg. Markowska et al., 2020) and disequilibrium effects in the cave are likely to play some role – though their roles have not yet been quantified. The exact control of d18O at Anhjohibe and Anjohikely is becoming the biggest unanswered question in stalagmite records from the region.

This is not an easy question to answer, there is no local IAEA site for precip d18O, the nearest weather station does not produce reliable enough data for climatological studies, and access to the site is limited by its remoteness, particularly during the wet season, which hinders cave monitoring. Effort is being made by multiple research groups. The commenter themselves is working on studies on both the d18O of precipitation, and the microclimate of Anjohibe. The lead author of this study is currently working on high-resolution records covering the observational era etc. Over the course of the next few years, the publication of these studies will shed new light on the interpretation and nuance of d18O in caves in northwest Madagascar.

As the top of AK1 does not extend into the observational era, this manuscript cannot shed more light onto the d18O mechanism without speculation. Section 5.1 (to be 5.2 in the revised manuscript) contains a discussion on d18O mechanisms with regards to the comparison of the similarities and differences with ANJ94-5, the coeval stalagmite. We believe this to be the limit of new information that can be gained from our data without speculation.

"About the cave: In an article published in 1997, Burney and colleagues described (p. 756) that Anjohikely is linked to Anjohibe with a subterranean passage (while citing Laumans et al., 1991). In addition to this, Burney et al. (1997) noted that "most passages in Anjohikely are relatively small in diameter, with somewhat limited development of speleothems (stalactites, stalagmites, and other dripstone formations). This makes me wonder about this sample from Anjohikely, particularly that its size is relatively big

if considering the former description and speleothem investigation by Burney et al."

We disagree with some aspects of the description provided by Burney et al on Anjohikely and stand-by our description of the cave in the third paragraph of section 5.1 (to be 5.2 in the revised version). Anjohibe certainly has a large number of very large stalagmites and flowstones, but in many places is reasonably bare of decoration. Anjohikely has far fewer large specimens but has a much greater percentage coverage of calcite. Burney et al 1997 is correct in stating that Anjohikely has tighter passages.

We have not been able to trace the Laumanns 1991 reference. We can only find Laumanns and Gebauer 1993. If the commenter has a copy, we would appreciate them sending it on.

"It may be helpful if the authors provide, at least, a skecth or map of the cave, with some illustrative figures of the cave entrance and the chamber from where the speleothems were extracted."

We have not seen a full survey map of Anjohikely. We believe that publishing an inaccurate sketch would not be informative.

"It would also be helpful to have a brief note about the overall microclimatic condition inside the cave (what is the air temperature inside, how about $pCO_2$ and relative humidity [RH])? Somewhere in the discussion (L. 224-229) that the authors discuss about kinetic fractionation as processes affecting stalagmites collected from Anjohibe (Wang et al., 20190) versus stalagmites from Anjohikely (their work). If the cave atmospheric exchange with its exterior atmosphere sounds important, information about cave microclimate (T, $pCO_2$, RH) could be crucial and discussed with more details. Also, I am far from believing that stalagmite growing in Anjohikely was precipitating in equilibrium with the cave drip water, as several studies have proven that almost none of the terrestrial carbonates precipitate in isotopic equilibrium with their drip water (Mickler et al., 2006; Tremaine et al., 2011; Day and Henderson, 2011; Deininger et al., 2012, Daëron et al., 2019)."

At no point in the manuscript do we suggest that our stalagmite was precipitated 100% in equilibrium, no stalagmite is. However, we have added a clause to explicitly state that the stalagmite did not grow under equilibrium conditions.

The comment is correct in stating that almost no terrestrial carbonates precipitate in isotopic equilibrium. The question is always 'how much disequilibrium fractionation is occurring', and 'is the magnitude of the disequilibrium effects on the stalagmite isotopes larger or smaller than the effects derived from climatic changes in d18O of precipitation'. Given the number and size of entrances of Anjohibe, Anjohikely and indeed all the caves in the region it is almost certain that AK1 and all regional stalagmites formed in some form of disequilibrium. The relative amount of which could be influenced by temperature, pCO2, and relatively humidity. There are numerous other potential influences on stalagmite d18O outside of disequilibrium vs equilibrium effects too, relating to storage, mixing, in-karst evaporation (e.g., Markowska 2020) and seasonal growth. As described above, there is a definite need for more detailed understanding of modern isotopic behavior in the region.

We regret that the sporadic nature of our visits to northwest Madagascar and the isolation of the area have prevented us from carrying out a full investigation into the cave microclimate. We did on, our last trip, install T and RH data-loggers at certain key sites, but do not yet have a record of significant length.

We consider these comments to be valid criticisms of this stalagmite, but also of the eight other stalagmite records published from the area. In our study, we clearly lay out which sections of the stalagmite (e.g., above 99mm) where we suspect disequilibrium effects may dominate the stable isotope record. We also provide in-depth discussion of how our stalagmite results compare to coeval stalagmite records and the likely causes of agreement and disagreement. We agree that there is always more that can be done in any stalagmite study to probe the proxy system. But, we believe our paper provides a reasonable discussion of various potential influences on stalagmite d18O, and the areas where we believe climatic processes dominate cave processes and vice-versa.

In contrast, other recently published stalagmites from the area show evidence of dis-equilibrium fractionation and a lack of replication without mention or discussion of the nuances of stalagmite d18O interpretation. For example: stalagmite ABC-1 (Li et al, Science Advances 2020), shows a dramatic, non-replicated, isotopic trend in a sta-lagmite from Anjohibe, which cannot be reasonably attributed to climatic change, and must be due to disequilibrium and drip hydrology effects. This was passed off as a cli-matic signal with no explanation of potential confounding effects. We have confidence that our stalagmite records climatic variability far more reliably than several recently published stalagmites from the area.

"About some statements in the paper: I disagree with the statement at Line 44 that "the impact of the 4.2 kyr event on the tropics and subtropics is unknown". âAËŸ c Please ′ consult Railsback et al., 2019, QSR, in which a review of the 4.2 ka event was pre-sented with a new isotopic record from Namibia. As the authors also make a compari-son with another stalagmite from Anjohibe about the 4.2 ka (Wang et al., 2019 QSR), this statement "unknown" needs to be revised. Other relevant information from India could be known by reading Kathayat et al. (2018, CP) and from China (Zhang et al., 2019 CP) Some comments in this section may be applicable to the preprint of the same authors (Scroxton, N., Burns, S. J., McGee, D., Godfrey, L. R.,Ranivoharimanana, L., and Faina, P.: Circum-Indian ocean hydroclimate at the mid to late Holocene transi-tion: The Double Drought hypothesis and consequences for the Harappan, Clim. Past Discuss. [preprint], https://doi.org/10.5194/cp-2020-138, in review, 2020)."

We agree this was an overgeneralization. There are a few papers that discuss the 4.2 kyr event, but nowhere near the number that discuss the event at temperate latitudes, see Kaniewski 2018, Bini 2019 and the Climate of the Past Special Issue in 2019. We have changed the sentence to: "In particular, the impact of the 4.2 kyr event on the tropics and subtropics is relatively unknown, particularly in the southern hemisphere, with only a few detailed studies, see Railsback et al., (2018)."

"I am also curious to know the opinion of the authors about the two replicated mid-

Holocene hiatuses reported in Voarintsoa et al. (2017 CP) in this region (one in Anjohibe, and another in Anjokipoty). As Anjohikely is a small cave, it is expected to behave like Anjokipoty (hence, should record the same hiatus of mid-Holocene deposition), and this comes back to my point earlier about the need of more details about the cave."

For the reader: The replicated hiatus in stalagmites ANJB-2 and MAJ-5 presented by Voraintsoa et al 2017 is between 7.8 and 1.6 kyr BP.

Wang et al. 2019 (QSR) has previously shown that the majority of this hiatus contains speleothem growth so there is no reason to suggest a multi-millennial scale drying of sufficient magnitude to stop speleothem growth. The presence of AK1 between 5 and 2 kyr BP confirms this. Of the hiatuses in stalagmite ANJ94-5 presented by Wang et al, the one at 4.2 kyr BP is replicated by AK1 and widely discussed in this paper. The 800 year hiatus around 6kyr BP and the 260 year hiatus around 8 kyr BP are both outside the growth interval of AK1 and therefore we cannot add to this discussion.

Anecdotally (and perhaps speculatively) we note that very few stalagmites from the area grew between the top of this stalagmite at 1.9 and the bottom of several stalagmites (ANJB-2, MAJ-5, AB2, AB3) around 1.7 kyr BP. This could indeed be indicative of a dry period in the region, and would explain the drying/progressive disequilibrium signal seen in AK1 from 2.3 kyr BP onwards. Therefore, our opinion is that dry periods lasting a few centuries may well be a regular feature of Holocene northwest Madagascar climate. Clearly further study is required from the stalagmites that do exist to determine the exact nature of this potential dry period, and other dry periods, in the region.

"Interpretation of the data In Figure 2: the authors indicate a hiatus within the palest color of the oldest generation of stalagmite. I wonder if this hiatus should actually be located at the boundary between the pale-brown color and the whiter color stalagmite (see annotated figure below), and if each of the color change throughout the sample

[Figure]

too can indicate other short term growth hiatus? As a matter of fact, only the bottom stalagmite has more age data. In contrast, the upper part of the stalagmite has lesser trenches. Why is that? Has there be any diagenesis at that brown-pale bottom part that allowed loss of U, hence the samples appear older?"

A hiatus at the change in color does indeed seem more likely at first glance and was indeed our working hypothesis for several months. However, 1) the U-Th age at 710mm is below the change in color, 2) there is consistency between growth rates below the hiatus at 707mm and above the hiatus until the U-Th date at 631mm, 3) Ages below the hiatus do not show consistently lower uranium concentrations than those above suggesting little or no uranium loss. There is no evidence of diagenesis. A higher position of the hiatus combined with the necessary drastic changes in growth rate is speculative and not supported by the data.

The presence of more U-Th ages below the hiatus occurred because we aim to achieve a consistent sampling of ages with respect to time, rather than with depth.

"At L. 150: Growth hiatus, what are/is the rational for saying "there is a growth hiatus"? I wonder what are the rational for saying that it indicates dry or wet conditions? While looking at the time series in Figure 3, it appears that the isotopic value bracketing the so-called hiatus are showing more negative values. Wouldn't this hiatus represent a wet condition? (May be an evaluation of the petrography (e.g., Railsback et al., 2013) would be useful here."

The rational for a growth hiatus is explained in the opening paragraph of section 5.1 (updated to section 5.2 in the revised version). Briefly, 1) AK1 shows a positive excursion into the hiatus not negative as suggested (although we do accept there is a negative excursion in the preceding few decades). 2) the hiatus is replicated in ANJ94-5 from nearby Anjohibe, which suggests the hiatus is driven by climate rather than a drip specific or cave specific change, 3) ANJ94-5 also shows a positive excursion into the hiatus, indicative of drying.

This comment is very similar to one made by Reviewer #1. We respect both of the commenters opinion that additional evidence is needed for a dry hiatus:

Upon reinspection of AK1 images. We can confirm that there are no truncated layers, a slight thinning on the stalagmite, an increase in d18O into the hiatus and a contraction crack (likely formerly aragonite) with little detrital material. We therefore believe this to be a Type L layer bounding surface, one caused by decreased precipitation. We have added this description to our results section and included the Railsback et al., 2013 reference. We believe that our manuscript is more robust and thank both the reviewer and Dr. Voarintsoa.

The new paragraph now reads: Between 4.30 and 3.84 kyr there is a growth hiatus. The layer bounding surface has no truncated layers, a slight thinning on the stalagmite, an increase in ïĄd'18O into the hiatus and a contraction crack (likely formed by the conversion of aragonite to calcite) with little detrital material. We interpret the layer bounding surface as Type L, one caused by decreased precipitation (Railsback et al., 2013). Further, the hiatus is replicated in stalagmite ANJ-94 from Anjohibe at (4.20– 3.99) (Wang et al., 2019b), also with a positive isotope excursion just prior, ruling out cave or drip specific drying. The replicated hiatus likely indicates dry conditions and potentially the driest conditions of the mid/late Holocene. The 4.2 kyr event therefore appears at least locally remarkable in northwest Madagascar. A dry anomaly is the opposite to the wet conditions recorded at 8.2 kyr BP (Voarintsoa et al., 2019), a Holocene climatic anomaly often viewed as a greater magnitude version of the 4.2 kyr event (Bond et al., 2001; Wang et al., 2013)

In addition, we are also returning to the stalagmite to see if a more detailed review of this layer bounding surface is necessary. We will report back as part of the formal response to the reviewer, but wanted to give this response more quickly, as part of the discussion phase.

"Presentation of the manuscript: I feel that the authors should clearly write a results

section and not combine results with discussion, or vice versa. Some short interpretations of the results are acceptable, if these are meant to emphasize the findings, but results should report results. With that said, In section 4.1 they should elaborate on the isotopic range, if there are any periodicity, highlight the extreme positive/negative excursion, and provide evidence of the hiatus. The author should also discuss about the growth rate. By looking at their Figure 3a, it seems that growth rate of the bottom part of the sample is slow vs. the upper part of the sample. I also feel that some information presented in the discussion belong to the results section (if not mentioning paragraph at L 230, L 235, and L. 242)"

This is a stylistic discussion. Our results section is organized in a result-interpretation format. Each paragraph begins with a result, and is followed by its interpretation: result 1, interpretation 1, result 2, interpretation 2, result 3, interpretation 3. We find this format to be less dry, more readable and less repetitive than: result1, result2, result3, interpretation1, interpretation2, interpretation3.

We agree that a description of the growth rate would be beneficial here and have included an extra sentence.

"The section about the regional variability in the African monsoon (Section 4.2) does not seem to belong into the results section."

Section 4.2 does not contain new results and therefore could be considered discussion. However, Section 4.2 is purely descriptive, and at no point interprets the data, and therefore could be considered results. Which section it belongs in is purely a stylistic distinction, but we have no issue with making this section 5.1 instead of 4.2 and have made the suggested change.

"Other detailed comments: At L 145: Can you please elaborate, or be specific, on the statement "change in drip hydrology" and "change in cave ventilation regime"?"

We have changed this paragraph to give more detail from the literature as to potential

specific causes of changes in drip hydrology.

"Figure 3: Can you please replace "Speleothem depth" with "distance from the top of the speleothem? In my understanding, depth is most commonly applied to sediments that are dig underground." Changed as suggested

"The authors mention in passing the diameter, the shape, and location of the drip axis (e.g., L. 245), it would be better to apply the layer-bounding surfaces approach (such approach was used in Wang et al., 2019 QSR) to quantify such changes. About the shape of the stalagmite again, I think there is quite a number of literature that they could use to back up their statement." We hope that the increased detail in the results section negates the need to alter this paragraph. We are not convinced that layer bounding surfaces approach constitutes quantification.

"Minor editorial errors: For some reasons, several of the in-text citations are replicated, if not only mentioning some at Lines 31, 36, 40, and 73). I guess some attention from the authors to avoid such replication is appreciated." Thank-you for pointing those errors out. Likely a result of the referencing software used. We have made the appropriate changes.

"If you use aragonitic, then calcitic seems to be parallel. However, it may be better to use aragonite and calcite (e.g., aragonite section..)" We agree, and have changed all aragonitic to aragonite

---

## Author Comment (AC2) · 5 Jan 2021

"Climate of the Past Submission cp-2020-137, "Possible expression of the 4.2 kyr event in Madagascar and the southeast African monsoon" by Scroxton et al., presents evidence from stalagmite AK1 from Anjohikely cave, northwest Madagascar and makes inferences about climate from 5000 to 2000 years BP. The most profound inference is of a "period of drought that lasted continuously from âĹij 4.32 and 3.83 ka BP". The authors have highlighted the hiatus recorded in the stalagmite AK1 between 4.32 and 3.83 kyr BP, replicating a hiatus in another stalagmite from nearby Anjohibe, and therefore indicating a significant drought around the time of the 4.2 kyr event in the region. The fundamentals of this research project are entirely based on the hiatus recorded in the stalagmites. The study draws on a stalagmite, AK-1, from Anjohikely cave, northwest Madagascar. I infer that the manuscript draws its scientific conclusions about the 4.2. ka event is from the hiatus recorded in the stalagmite AK-1, and comparisons with the previously published studies. Suggestion 1. I think it should be, hence, provide the explicit petrographic studies of the hiatus. The layer bounding studies as discussed by Railsback et al., 2013 is important for this project because it will provide robust evidence demonstrating if the periods of non-deposition, either because of exceptionally wet or dry conditions. Reference : Railsback, L.B., Akers, P.D., Wang, L., Holdridge, G.A., Voarintsoa, N., 2013. Layer-bounding surfaces in stalagmites as keys to better paleoclimatological histories and chronologies. Int. J. Speleol. 42, 167–180."

The rational for a growth hiatus is explained in the opening paragraph of section 5.1 (updated to section 5.2 in the revised version). Briefly, 1) AK1 shows a positive excursion into the hiatus typical of a drying drip. 2) the hiatus is replicated in ANJ94-5 from nearby Anjohibe, which suggests the hiatus is driven by climate rather than a drip specific or cave specific change, 3) ANJ94-5 also shows a positive excursion into the hiatus, indicative of drying.

This comment is very similar to one made in the public comment by Ny Riavo G. Voarintsoa. We respect both of the commenters opinion that additional evidence is needed for a dry hiatus:

Upon reinspection of AK1 images. We can confirm that there are no truncated layers, a slight thinning on the stalagmite, an increase in d18O into the hiatus and a contraction crack (likely formerly aragonite) with little detrital material. We therefore believe this to be a Type L layer bounding surface, one caused by decreased precipitation. We have added this description to our results section and included the Railsback et al., 2013 reference. We believe that our manuscript is more robust and thank both the reviewer and Dr. Voarintsoa.

The paragraph now reads: Between 4.30 and 3.84 kyr there is a growth hiatus. The layer bounding surface has no truncated layers, a slight thinning on the stalagmite, an

increase in ïĄď18O into the hiatus and a contraction crack (likely formed by the conversion of aragonite to calcite) with little detrital material. We interpret the layer bounding surface as Type L, one caused by decreased precipitation (Railsback et al., 2013). Further, the hiatus is replicated in stalagmite ANJ-94 from Anjohibe at (4.20–3.99) (Wang et al., 2019b), also with a positive isotope excursion just prior, ruling out cave or drip specific drying. The replicated hiatus likely indicates dry conditions and potentially the driest conditions of the mid/late Holocene. The 4.2 kyr event therefore appears at least locally remarkable in northwest Madagascar. A dry anomaly is the opposite to the wet conditions recorded at 8.2 kyr BP (Voarintsoa et al., 2019), a Holocene climatic anomaly often viewed as a greater magnitude version of the 4.2 kyr event (Bond et al., 2001; Wang et al., 2013)

In addition, we are also returning to the stalagmite to see if a more detailed review of this layer bounding surface is necessary. We will report back as part of the formal response to the reviewer, but wanted to give this response more quickly, as part of the discussion phase.

––––––––––––––––––––––––

---

## Referee Comment (RC2) · Anonymous Referee #2 · 6 Jan 2021

There are still a lot of discussions surrounding the infamous 4.2-kyr event as its timing, nature and spatial extent remain uncertain, not to mention the lack of evident forcing mechanisms (e.g. solar, volcanic, AMOC. . .). New precisely dated records are therefore needed to reduce the current uncertainties, particularly from the tropics. The new stalagmite record from Anjohikely Cave in northern Madagascar shows a hiatus between 4.32 and 3.83 kyr BP and thus confirms a hiatus in a stalagmite from Anjohibe Cave approx. km away from Anjohikely. However, the hiatus in stalagmite ANJ94-5 from Anjohibe Cave lasted from 4.2 to 4.0 kyr BP which is most likely related to slightly different age models, but not really covered in the manuscript. In my opinion, the complete lack of an adequate discussion of the age models and the resultant timing of the 4.2 kyr event is one of the main weaknesses of this manuscript and major revisions

are required to address this crucial aspect. There is quite a lengthy discussion on the isotope profiles which seems to be slightly disconnected from the main aspect (the climate-induced hiatus) of the manuscript. The authors should highlight the positive shifts at the onset of the 4.2 kyr event more effectively in order to document the abruptness of the 4.2 kyr event in greater detail. While we are here, it is quite interesting that stalagmite ANJ94-5 (Wang et al., 2019) is showing a positive shift in del18O whereas such a comparable shift is missing in the AK1 profile (Fig. 5). What is the reason for this mismatch? Is there a possibility to increase the sampling resolution of the AK1 isotope profile to identify such a comparable isotopic towards the 4.2 kyr event? Furthermore, a better documentation (thin section or macro images) of the hiatus would make the manuscript much stronger. As mentioned above, a detailed discussion about the onset of the 4.2 kyr event should be a central aspect of the paper, which would include a detailed discussion about the uncertainties of the stalagmite age models (AK1 and ANJ94-5) which are based on different approaches to develop an age model: the age model for stalagmite AK1 is based on OxCal and the one for ANJ94-5 is based on StalAge. I would recommend to use the same age modelling approach and to cite the 2-sigma uncertainties for the ages of the hiatus throughout the manuscript. The authors should also add a more detailed discussion of the chronologies of other hydroclimate records from southeast Africa (e.g., Lake Malawi, Lake Masoko) as the timing of the onset of drought conditions is crucial. The onset of drought conditions at Lake Masoko began at 4.5 kyr, at 4.4 kyr at Lake Malawi, 4.3 at Anjohibe and 4.5 kyr (see chapter 5.2). The authors comment on this in only one (incomplete) sentence "The age errors for most records are around ±600 years (2-sigma) for the stalagmite records and ±200 years (2ïĄş) for most other records". Their conclusion that the hydroclimate anomalies in these records is synchronous with the 4.2 kyr event is therefore too optimistic and not really supported by all of the records. Chapter 5.3 "Timing of the middle to late Holocene climate shifts in the SEAfM" must be therefore expanded to document the chronologies of the key-records in much greater detail. Overall, a proper evaluation of the chronologies of the different records is required.

---

## Author Comment (AC3) · 17 Jan 2021

We would like to thank the reviewer for their contribution. They have raised some interesting points which certainly require discussion, and in many cases, the addition of more detail and more nuance to our manuscript, which we feel has improved the manuscript.

"There are still a lot of discussions surrounding the infamous 4.2-kyr event as its timing, nature and spatial extent remain uncertain, not to mention the lack of evident forcing mechanisms (e.g. solar, volcanic, AMOC. . .). New precisely dated records are therefore needed to reduce the current uncertainties, particularly from the tropics. The new stalagmite record from Anjohikely Cave in northern Madagascar shows a hiatus between 4.32 and 3.83 kyr BP and thus confirms a hiatus in a stalagmite from Anjohibe Cave approx. km away from Anjohikely. However, the hiatus in stalagmite ANJ94-5 from Anjohibe Cave lasted from 4.2 to 4.0 kyr BP which is most likely related to slightly different age models, but not really covered in the manuscript. In my opinion, the complete lack of an adequate discussion of the age models and the resultant timing of the 4.2 kyr event is one of the main weaknesses of this manuscript and major revisions are required to address this crucial aspect. There is guite a lengthy discussion on the isotope profiles which seems to be slightly disconnected from the main aspect (the climate-induced hiatus) of the manuscript. The authors should highlight the positive shifts at the onset of the 4.2 kyr event more effectively in order to document the abruptness of the 4.2 kyr event in greater detail. While we are here, it is guite interesting that stalagmite ANJ94-5 (Wang et al., 2019) is showing a positive shift in del180 whereas such a comparable shift is missing in the AK1 profile (Fig. 5). What is the reason for this mismatch? Is there a possibility to increase the sampling resolution of the AK1 isotope profile to identify such a comparable isotopic towards the 4.2 kyr event? As mentioned above, a detailed discussion about the onset of the 4.2 kyr event should be a central aspect of the paper, which would include a detailed discussion about the uncertainties of the stalagmite age models (AK1 and ANJ94-5) which are based on different approaches to develop an age model: the age model for stalagmite AK1 is based on OxCal and the one for ANJ94-5 is based on StalAge. I would recommend to use the same age modelling approach and to cite the 2-sigma uncertainties for the ages of the hiatus throughout the manuscript."

As suggested by the reviewer, the offset age between the onset of the hiatus in AK1 and ANJ94-5 can certainly be treated in more detail, and we are happy to provide more information on what is the key conclusion of our manuscript, and we have made several changes to section 5.2.

We believe the offset in the hiatus starting point is likely a combination of two factors: age model discrepancies and drip hydrology. The last stable isotope datapoint of
ANJ94-5 is 4199 yrs at 520mm. Age uncertainties are not supplied with the ANJ94-5 data but are likely of the order of +-30yrs, which is the 2s uncertainty at the nearest age at 535mm. StalAge has a constant growth-rate bias, and the ANJ94-5 age model is 40 years too young at the nearest age tie point, outside the 2sigma uncertainty. This suggests the ANJ94-5 age model may run too young at the hiatus also by around forty years. In comparison, the last stable isotope datapoint of AK1 is 4307yrs at 707.25mm. The OxCal age uncertainty is +-60 years at the data-point, but is likely comparable to the ANJ94-5 uncertainty given the nearest U-Th tie point at 710.5mm is 4325yrs +-34. OxCal has its own mean growth rate bias when extrapolating beyond tie-points to hiatuses. We calculate the effect of this on the age of the last stable isotope point to be 5 years (too old). Therefore, a supposed 90 year difference in hiatus may only be half that.

However, the age model discrepancies may be irrelevant when considering that the two stalagmites have different drip hydrologies, and therefore are likely to dry out at different rates. (This is why re-running both stalagmite age models with the same software would not necessarily be informative). The exact timing of a hiatus is unique to each stalagmite. The temporal resolution of AK1 is 4-5x that of ANJ94-5, and the sampling resolution is 6x so an higher resolution pre-excursion transect of AK1 would likely not include any additional information. We are happy to clarify this point in the manuscript too as it contributes to our discussion on drip hydology differences.

If the hiatus is idosyncratic, then perhaps instead we should be looking at the drying before the hiatus. For example, we can look at the last low d18O value before the hiatus. In ANJ94-5 the last low d18O value below -5 per mill before the dramatic drying trend is at 4295yrs at 536mm. This is just 12 years different to the hiatus of AK1, which given a possible young bias in the ANJ94-5 age model and a sampling resolution of 23 years could be considered simultaneous. Alternatively, we could choose the last d18O minima in both stalagmites. In ANJ94-5 the last d18O value below -6 per mill, which appears to be a more natural peak, occurs at 4337 yrs (543mm). In AK1, the d18O

CPD
minima prior to the hiatus occurs at 4321 +-46 years at 708.75mm. Again, this can be considered simultaneous. Can a unified age be established here? 4320 yr BP +-50 years seems reasonable. But the issue here is whether the exact choice of location is subjective, and due to the proximity of the hiatus, a statisitical fit is likely to fail without sufficient post change data points.

Instead, it might be better to look at broader centennial scale trends rather than isolate individual stable isotope points. Both stalagmites show wetting signals beween 4.7 and 4.6 kyr BP, and a drying from 4.5 kyr onwards.

All of this analysis is indeed possible. The question we also have to ask is whether this analysis is robust and within the bounds of reasonable interpretation of the data. We could easily imagine presenting this analysis and a different reviewer telling us that we were overreaching the data, given the dating resolution, the sampling resolution and the idiosyncrasies of drip hydrology. We hope we have struck the right balance in our revisions.

"Furthermore, a better documentation (thin section or macro images) of the hiatus would make the manuscript much stronger."

In light of the comment from Dr. Voarintsoa, and the review from anonymous reviewer 1, we think this is a good suggestion for providing extra information on the hiatus.

"The authors should also add a more detailed discussion of the chronologies of other hydroclimate records from southeast Africa (e.g., Lake Malawi, Lake Masoko) as the timing of the onset of drought conditions is crucial. The onset of drought conditions at Lake Masoko began at 4.5 kyr, at 4.4 kyr at Lake Malawi, 4.3 at Anjohibe and 4.5 kyr (see chapter 5.2). The authors comment on this in only one (incomplete) sentence "The age errors for most records are around  $\pm 600$  years (2-sigma) for the stalagmite records and  $\pm 200$  years (2ï EZA Âÿs) for most other records". Their conclusion that the hydroclimate anomalies in these records is synchronous with the 4.2 kyr event is therefore too optimistic and not really supported by all of the records.Chapter 5.3

CPD
"Timing of the middle to late Holocene climate shifts in the SEAfM" must be therefore expanded to document the chronologies of the key-records in much greater detail. Overall, a proper evaluation of the chronologies of the different records is required."

We agree with the reviewer comment that calling these records synchronous with the 4.2 kyr event is optimistic. This was our aim, as we discuss in section 5.4. However, it is clear that we did not make this as obvious as it could/should be. Therefore, we agree that to make this clearer, additional discussion is needed. Firstly, we have removed reference to the 4.2kyr event from section 5.3, and the first paragraph of the conclusion, to discuss what the records show without the bias of interpretation. Coincidence with the 4.2 kyr event is discussed exclusively in section 5.4.

Discussion of the age errors of each individual record is complicated. Most studies do not discuss age model error, or in many cases even calculate them. Mostly we have to rely on the age determination errors, which are frequently uncalibrated radiocarbon ages, and are of course smaller than age errors on interpolated ages. Nether-theless, we agree it is important to state these age errors in the manuscript. The large errors make correlating events difficult or easy depending on your proclivity. They are mostly within 2sigma range of each other, and mostly (but not completely) in range of the 2sigma 4.26 kyr event. Alternatively, you could view these errors as being large enough to be unable to reliably prove synchronicity. As 2 sigma age errors are not cut-off boundaries of yes/no and are in fact arbitrary markers on a distribution where the mean/median (depending on whether they are Gaussian or not) is the most likely. Determining age model uncertainty resolved synchronicity is beyond the scope of this paper. Indeed, it was part of the deliberate choice in the separation of this paper with its co-submission, which does deal with the highest resolution, precisely dated records in the region, with publicly archived interpolated age uncertainty data. We believe it is more informative here to continue with the mean ages. We hope that the rephrasing of several sentences throughout sections 5.3 and 5.4 help better state this inherent uncertainty in the analysis.
**Attached is the new Figure 4**

Figure 4 caption: Close-up of the mid- to late- Holocene hiatus in AK1. a) image with no annotation. b) image with annotation. Red shaded areas denote U-Th date sampling locations, blue shaded areas denote stable isotope transect trench, visible laminae in thin black lines, the mid- to late- Holocene hiatus in thick black line, unmarked circular pits are stable isotope drill holes from a low resolution pilot study. c) higher zoom unannotated image of the mid- to late- Holocene hiatus.

**CPD**
Fig. 1.

---

## Author Comment (AC4) · 9 Feb 2021

We would like to thank the two anonymous reviewers, and Dr. Voarintsoa for their discussion. We believe the manuscript has been improved via their input. In this final report we provide a summary of the main discussion points raised by the reviewers, either individually or together. Individual detailed responses have already been submitted as part of the discussion phase.

The main issue raised by all three reviewers was that the evidence provided for the hiatus in stalagmite AK1 was insufficient to prove a dry event. In the majority of stalagmite studies we have seen, a hiatus that replicates between two different caves and which shows a positive isotope excursion leading into the event would be considered

sufficient evidence of a dry event. And while 'wet' hiatuses are possible, they seem very rare in the literature. However, as pointed out, this hiatus is not just part of the record, it is the central result of this manuscript. Therefore, we agree with all three reviewers that the hiatus warrants extra scrutiny. As suggested, we have investigated the stalagmite in light of the layer-bounding surfaces framework of Railsback et al. (2013) and determined the hiatus to be a Type L bounding surface, one likely caused by dry conditions. The evidence for this is an absence of truncated layers, a slight thinning of layers and narrowing of the stalagmite, an increase in d18O into the hiatus, and little detrital material. The third paragraph of section 4.1 is expanded to include this new information. A small fourth paragraph is included to briefly discuss a second Type L bounding surface at 694mm, as suggested by Dr. Voarintsoa. In our response to reviewer 1 we erroneously stated that there might be a potential contraction crack from the recrystallization of former aragonite – we no longer believe this to be the case, unless selective recrystallisation occurred at the hiatus but not the rest of the aragonite stalagmite, which would be highly speculative. We also include a new figure four which shows three close-up images of the hiatus, one annotated.

Reviewer 2 asked us to go further with the description of the hiatus and include a discussion on the relative roles of age model uncertainty and age model choice in the difference in timing of the hiatus between AK1 and ANJ94-5, the replicating stalagmite from nearby Anjohibe. We agree that age model uncertainty plays a role in the difference, as does age model choice. However, likely of equal or greater importance is the drip hydrology of the two stalagmites. The onset of drying (positive d18O excursion) in the two stalagmites are much closer in age than the physical changes (hiatus onset). This suggests that the exact timing of the hiatus onset is determined by the size of the karst water store and drip hydrology. We are happy to include this discussion, but we caution that interpreting age model differences less than the error bounds of the stalagmites could be regarded by other reviewers as over-interpretation. We hope we have found the correct balance between nuanced discussion and avoiding over-interpretation. The first half of section 5.2 has been expanded to include the discussion on this topic. We have, as suggested, placed it ahead of the discussion on the isotopic similarities between the two records.

We disagree with reviewer 2 that the stalagmite age models need to be re-run using the same software. Our choice of age model is based on the U-Th age profile, frequency of dating and known biases of the nine (or so) age modelling software packages available to the stalagmite community. In our opinion it is more prudent to discuss the biases of the age models of the two stalagmites, than it is to run both stalagmites using the same age model and therefore come up with a more similar answer, but one in which the bias is not removed.

Reviewer 2 also asked for more detailed discussion of the chronologies of the other records, saying "that the hydroclimate anomalies in these records is synchronous with the 4.2 kyr event is therefore too optimistic and not really supported by all of the records". We agree with the Reviewer 2 here. Acknowledging that there is doubt over the synchronicity with the 4.2 kyr event was our aim. Therefore, we needed to state this more clearly, and introduce a more thorough discussion of the age model errors of other records. This is not a straightforward task, as most studies no not discuss age model error, or even calculate error at interpolated data-points. Mostly we have to rely on the age determination errors, which are frequently uncalibrated radiocarbon ages, and are of course smaller than age errors on interpolated ages. A full discussion on age model uncertainty resolved synchronicity is beyond the scope of this manuscript. In fact, it is the scope of our companion manuscript cp-2020-138.

We have included a more thorough description of the age uncertainties in the other records at the start of section 5.3 and rephrased several sentences in section 5.4 to add to the discussion over age uncertainty and synchronicity with the 4.2kyr event. We also decided to make a stylistic change to the entire manuscript based on this comment. We have decided to remove "the 4.2 kyr event" as a phrase throughout and restrict its discussion to section 5.4. We have replaced "4.2 kyr event" with "mid- to late- Holocene transition". This avoids the linking of the two events until it is ready to

be discussed in full.

Dr. Voarintsoa asked us about the interpretation of the stable oxygen isotope record of AK1, and the potential factors that drive oxygen isotope variability inside the cave, including disequilibrium fractionation. We agree with this line of questioning in general terms but feel it is a more general question that can be asked of all records published so far from the region. As this discussion applies to all stalagmites from the region, we discuss this in section 5.1 rather than 4.1.

Our interpretation follows that of previous studies. The highly seasonal nature of rainfall at the site, the single source area of precipitation and the site's proximity to the ocean suggest that the amount effect likely dominates the d18O of rainfall. Given the aridity of the region and the openness of the cave, processes such as evaporative enrichment in the karst and disequilibrium fractionation during carbonate precipitation all likely play a role in signal modification. We believe we are quite open about these multiple potential processes but disagree with the assertion that we state our stalagmite grew in perfect equilibrium. This is erroneous, we do not make this claim. It is correct that disequilibrium fractionation is likely a component of all stalagmite precipitation, but it is the amount of disequilibrium fractionation that is important, and further, the amount of disequilibrium fractionation is likely driven by climatic processes.

Fully reconciling the different influences on stalagmite d18O at this site is beyond the scope of this paper, but we agree that a better understanding of the exact controls on d18O at this site is fast becoming a significant issue in stalagmite records from the region. We know that it is an area of active research, both by our group, and the work of Dr. Voarintsoa and we look forward to the publication of these studies in the coming years.

We have responded to the more minor comments from the reviewers in the individual responses, and thank them once again for their input.